

# Limits to load-lifting performance in a passerine bird: the effects of intraspecific variation in morphological and kinematic parameters

Yang Wang[1,*], Yuan Yin[1,*], Shiyong Ge[1], Mo Li[1], Qian Zhang[1], Juyong Li[1], Yuefeng Wu[1], Dongming Li[1] and Robert Dudley[2]

[1] Key Laboratory of Animal Physiology, Biochemistry and Molecular Biology of Hebei Province, College of Life Sciences, Hebei Normal University, Shijiazhuang, Hebei, China
[2] Department of Integrative Biology, University of California, Berkeley, CA, USA
* These authors contributed equally to this work.

## ABSTRACT

Although more massive flight muscles along with larger wings, higher wingbeat frequencies and greater stroke amplitudes enhance force and power production in flapping flight, the extent to which these parameters may be correlated with other morphological features relevant to flight physiology and biomechanics remains unclear. Intraspecifically, we hypothesized that greater vertical load-lifting capacity would correlate with higher wingbeat frequencies and relatively more massive flight muscles, along with relatively bigger hearts, lungs, and stomachs to enhance metabolic capacity and energy supply, but also with smaller body size given the overall negative allometric dependence of maximum flight performance in volant taxa. To explore intraspecific correlates of flight performance, we assembled a large dataset that included 13 morphological and kinematic variables for a non-migratory passerine, the Eurasian tree sparrow (*Passer montanus*). We found that heavier flight muscles and larger wings, heavier stomachs and shorter bills were the most important correlates of maximum load-lifting capacity. Surprisingly, wingbeat frequency, wing stroke amplitude and masses of the heart, lungs and digestive organs (except for the stomach) were non-significant predictor variables relative to lifting capacity. The best-fit structural equation model (SEM) indicated that load-lifting capacity was positively correlated with flight muscle mass, wing area and stomach mass, but was negatively correlated with bill length. Characterization of individual variability in flight performance in a free-ranging passerine indicates the subtlety of interaction effects among morphological features, some of which differ from those that have been identified interspecifically for maximum flight performance in birds.

## INTRODUCTION

Birds exhibit a broad diversity of flight-related morphological and physiological characteristics (*Hedenström, 2002*; *Lee et al., 2014*; *Puttick, Thomas & Benton, 2014*; *Altshuler et al., 2015*; *Butler, 2016*), many of which reflect multiple trade-offs in flight

Corresponding author
Dongming Li, lidngmng@gmail.com

performance (*Goslow, Dial & Jenkins, 1990*; *Ellington, 1991*; *Chai & Dudley, 1999*; *Lind, 2001*; *Lind & Jakobsson, 2001*; *Altshuler et al., 2015*). For example, larger species typically possess bigger wings and higher pectoral muscle mass, whereas wingbeat frequency declines with increasing body mass (*Groom et al., 2018*); higher wingbeat frequencies and greater stroke amplitudes nonetheless yield increased force and power production (*Chai & Dudley, 1995*; *Hedenström, 2002*; *Altshuler et al., 2015*). Wing morphological and kinematic features along with flight-related muscle, are thus key variables influencing flight performance.

Avian flight is an energy-demanding activity requiring powerful respiratory and cardiovascular systems to support the intense metabolism of the associated skeletal muscles (*Hedenström, 2002*; *Lee et al., 2014*; *Altshuler et al., 2015*; *Butler, 2016*; *Nespolo et al., 2018*). As major metabolic engines, heart and lung capacities underpin both burst power and endurance flight performance (*Bishop & Butler, 1995*; *Wright, Gregory & Witt, 2014*). Moreover, larger nutritional organs and increases in the quantity of digestive enzymes and nutrient transporters are essential to meet the high-energy demands of flight (*Karasov & McWilliams, 2005*). For example, many birds exhibited a reduction in the pectoralis primary mass as a consequence of lowered nutritional supplies in post-migratory periods (*Dietz et al., 2007*; *Piersma & Dietz, 2007*). Therefore, respiratory, cardiovascular and nutritional systems represent important features for sustaining powered flight.

Other morphological factors contributing to an increase in body mass could be considered as hindering features that reduce available power for flight (*Ellington, 1991*; *Dial, 2003*). For example, a toothless beak in extant birds is believed to increase flight efficiency by reducing overall body mass (*Louchart & Viriot, 2011*). However, relationships among diverse morphological features relative to flight capability in free-living birds have not been well investigated.

Maximum load-lifting capacity (as imposed via asymptotic loading; *Buchwald & Dudley, 2010*), is an informative means of evaluating burst capacity in volant taxa (*Altshuler, Dudley & McGuire, 2004*; *Altshuler et al., 2010*; *Sun et al., 2016*). To date, interspecific comparisons have evaluated morphological and functional correlates of the maximum load-lifting capacity of free-living birds (*Hedenström, 2002*; *Lee et al., 2014*; *Altshuler et al., 2015*; *Butler, 2016*; *Groom et al., 2018*; *Nespolo et al., 2018*); however, much less information is available about intraspecific determinants of maximum loading-lifting capacity. Intraspecifically, we hypothesize that maximum load-lifting capacity will be positively influenced by key morphological and kinematic features relevant to force and power production (e.g., relatives size of flight muscles and the wings, wingbeat frequency and stroke amplitude), as well as by metabolically relevant features (including hearts, lungs and digestive organs relevant to sustained flight).

To evaluate these hypotheses, we assembled a large dataset with a total of 13 variables of morphological, internal anatomical, and biomechanical variables for a non-migrant passerine, namely the Eurasian tree sparrow (*Passer montanus*). We first determined statistically the most important variables contributing to maximum load-lifting capacity (total lifted load, i.e., the sum of body mass and the maximum supplemental load), as

determined by asymptotic load-lifting experiments. We further assessed the relative contribution of each variable to maximum load-lifting capacity, then identified the most critical factors influencing intraspecific variation using multiple-variable interactions and structural equation model (SEM) in this transient feature of flight performance.

## MATERIALS AND METHODS

### Bird collection

Totals of 33 male and 39 female adult Eurasian tree sparrows were captured opportunistically using mist nets during the late winter of 2017 (i.e., 13 March to 1 April) at five different lowland sites (Dongyangshi: N37.9667°, E114.602°; Yuhuaqu: N38.021°, E114.526°; Changanqu: N38.058°, E114.547°; Xiangzigou: N38.309°, E114.001°; Mayu: N38.322°, E113.962°; site elevational range from 80 m to 203 m) around Shijiazhuang City, Hebei Province, People's Republic of China. Within 30 min post-capture, body mass to within 0.01 g was measured for each bird using a portable digital balance.

### Load-lifting assay

Birds were placed individually (within 3 h of capture) in a rectangular flight chamber (45 cm × 45 cm × 150 cm) made from transparent acrylic sheet, as used in previous experiments (*Sun et al., 2016*). Each bird was evaluated for asymptotic load-lifting capacity using an assay described in detail elsewhere (*Sun et al., 2016*). Briefly, a thread with different plastic beads (each approximately 1.0 g in mass) and positioned at fixed linear intervals was attached to the left tarsometatarsus of the sparrow. When released from the bottom of the chamber, birds typically flew vertically towards the top, asymptotically lifting more and more beads until a maximum load was attained. Two cameras were used in this experiment; one high-speed video camera (JVC GCP100BAC; operated at 50 frames$^{-1}$; see Supplemental Material) positioned laterally at a distance of 80 cm to the chamber was used to film the beads remaining on the chamber floor during maximum load-lifting flight and thus by subtraction to determine the total extra weight lifted by the bird. The other synchronized camera (JVC GCP100BAC, operated at 250 frames$^{-1}$; see Supplemental Material), was positioned laterally near the top of the chamber (*Sun et al., 2016*) and was used to obtain wingbeat frequency and stroke amplitude.

Multiple ascending flights were recorded for each bird (mean of 4.7 flights), and the maximum weight lifted within the series was assumed to indicate the limit to load-lifting flight performance. Wingbeat frequency was determined from the number of frames required to complete an integral number of wingbeats for a composite sequence containing multiple flapping cycles, but starting and finishing at the same vertical position of the wings. Wing stroke amplitude was calculated as the angle between extreme wing tip positions (i.e., the point of the outermost primary feather relative to the longitudinal body axis) at extremes of the nominally vertical wing stroke, as filmed by the top lateral camera (*Sun et al., 2016*). A mean value for stroke amplitude was calculated from three to five separate measurements within each bout of maximum load within the final 0.5 s of peak lifting.

## Morphological, anatomical and kinematic parameters

To identify potential influences total lifted load, we determined for each individual bird a total of 13 variables of morphological (sex, body mass, bill length, total wing area, aspect ratio), internal anatomical (the masses of flight muscle, heart, lung, liver, stomach and gut length) and biomechanical relevance (wingbeat frequency and amplitude). Following load-lifting experiments, each bird was immediately euthanized with phenobarbitone (7.5 µl g$^{-1}$ body mass), and its bill length was measured to the nearest 0.1 mm using Vernier calipers. The right wing was photographed for measurements of wing area and wing length $R$ (analyzed using ImageJ, National Institutes of Health, Bethesda, MD, USA); total wing area $S$ is given by twice the area of the right wing. The aspect ratio is given by $(2R)^2/S$. The pectoralis major, pectoralis minor and the whole heart, lung, liver (all following blotting to remove blood), along with the fresh gut and stomach (food residue was removed by washing with water), were then excised and weighed using a digital balance sensitive to 0.1 mg (with gut length measured to +1 mm). All protocols were approved by the Ethics and Animal Welfare Committee (No. 2013-6) and by the Institutional Animal Care and Use Committee (HEBTU2013-7) of Hebei Normal University, China and were carried out under the auspices of scientific collecting permits issued by the Department of Wildlife Conservation (Forestry Bureau) of Hebei Province, China.

## Statistical analyses

We calculated means and standard deviations for gross morphological (body mass, bill length, total wing area, aspect ratio), internal anatomical (the masses of flight muscle, heart, lung, liver, stomach, and gut length) and kinematic variables (wingbeat frequency and wing stroke amplitude). We determined Pearson correlations between maximum load-lifting flight performance (i.e., total lifted load) and all other variables, including sex as a discrete covariate. We then implemented a generalized linear model using the *glm* function in Program R v. 3.4.2 (*Pinheiro et al., 2015*) to model relationships between dependent factors (i.e., total lifted load) and all independent variables, with sex as a discrete covariate. All continuous variables were scaled (i.e., centralized and standardized; *Schielzeth, 2010*) before such modeling to reduce multicollinearity. Multiparameter models were discarded if a nested model (i.e., collinearity among factors) containing a subset of the same parameters had a better Akaike's Information Criterion (AIC) score. To account for model selection uncertainty, model-averaged estimates of variable coefficients were computed using the "best model set," defined as the set of models for which delta AIC was less than six (*Burnham & Anderson, 2002*). All possible models were averaged to identify the most important variables, using the importance score in the MuMIn package (*Kamil, 2013*) of R v.3.4.2. We ranked all variables selected by the average model and then considered those variables with a higher relative importance score (i.e., >0.7) as determinant variables for total lifted load. We further assessed whether there were level-two interaction effects on maximum load-lifting flight performance and then selected those most important variables underlying variance in total lifted load by identifying those with relative importance scores >0.7. Finally, we constructed a SEM in

**Table 1 Measured variables for morphology, internal anatomy, kinematics, and load-lifting performance for Eurasian tree sparrows (*Passer montanus*) as averaged for the two sexes, and their correlations with total lifted load (Lifted mass + body mass).**

| Category of variable | Variable | Mean | SD | Correlation with total lifted load | |
|---|---|---|---|---|---|
| | | | | *r* | *r*$^2$ |
| Morphology | Body mass (g) | 19.216 | 1.194 | 0.539*** | 0.291 |
| | Bill length (mm) | 8.558 | 0.497 | −0.051 | 0.003 |
| | Wing area (cm$^2$) | 80.678 | 6.581 | 0.362** | 0.131 |
| | Aspect ratio | 2.229 | 0.211 | −0.094 | 0.009 |
| Internal anatomy | Flight muscle mass (g) | 2.450 | 0.250 | 0.669*** | 0.448 |
| | Heart mass (g) | 0.187 | 0.026 | 0.376** | 0.141 |
| | Lung mass (g) | 0.097 | 0.017 | 0.258* | 0.067 |
| | Liver mass (g) | 0.345 | 0.061 | 0.264* | 0.07 |
| | Stomach mass (g) | 0.323 | 0.075 | 0.325** | 0.106 |
| | Gut length (mm) | 136.395 | 7.297 | 0.262* | 0.069 |
| Kinematics | Wingbeat frequency (Hz) | 5.154 | 0.486 | 0.171 | 0.029 |
| | Wing stroke amplitude (°) | 151.389 | 3.704 | 0.079 | 0.006 |
| Load-lifting performance | Maximum load (g) | 26.027 | 4.592 | | |
| | Total lifted load (g) | 45.243 | 5.124 | | |

**Note:**
Asterisk represents significant correlation between each variable and total lifted load. *, $P < 0.05$; **, $P < 0.01$; ***, $P < 0.001$.

the *lavaan* package (*Rosseel, 2012*) of Program R v.3.4.2, including all combinations of those important variables as identified by AIC scores and selected the best model with a chi-square test, the root mean square error of approximation (RMSEA), the standard root mean square residual (SRMR), and the comparative fit index (CFI). We assumed a well-fitted model to have a *p*-value >0.05 for the chi-square test, a RMSEA and SRMR test with values less than 0.1 and a CFI close to 1 (i.e., >0.9). The relationships between measured variables and total lifted load were represented by regression coefficients; all path coefficients used standardized estimates.

## RESULTS

Total lifted load was positively correlated with diverse traits, including body mass, wing area, the masses of flight muscle, heart, lung, liver, stomach and length of gut (Table 1). Among all measured variables, variation in total lifted load was best explained by variability in bill length, stomach mass, gut length, wing area and flight muscle mass (Table 2). However, sex and other morphological (body mass, aspect ratio), internal anatomical (masses of heart, lung, and liver), and kinematic features (wingbeat frequency and wing stroke amplitude) did not significantly predict variation in maximum load-lifting capacity (Table 2). Specifically, heavier flight muscles, a greater stomach mass, larger wings and a longer gut but also shorter bills were strongly correlated with total lifted load. Among these five variables, flight muscle mass, bill length, wing area and stomach mass were the four most important factors predicting total lifted load when all level-two interactions were considered (Table 3).

**Table 2 Model-averaged statistical results in the best model set (delta AIC < 6) correlating total lifted load by Eurasian tree sparrows (*Passer montanus*) with morphological, physiological, and kinematic parameters.**

| Response variable | Estimate | Adjusted SE | 95% CI | Relative importance |
|---|---|---|---|---|
| (Intercept) | 0.006 | 0.095 | −0.180, 0.191 | |
| Bill length | −0.259 | 0.090 | −0.436, −0.083 | **1.00** |
| Flight muscle mass | 0.579 | 0.130 | 0.324, 0.834 | **1.00** |
| Stomach mass | 0.252 | 0.100 | −0.057, 0.447 | **1.00** |
| Wing area | 0.262 | 0.133 | 0.001, 0.523 | **0.96** |
| Gut length | 0.148 | 0.086 | −0.021, 0.317 | **0.70** |
| Heart mass | −0.136 | 0.108 | −0.349, 0.076 | 0.48 |
| Wingbeat frequency | 0.105 | 0.088 | −0.067, 0.277 | 0.46 |
| Aspect ratio | 0.146 | 0.140 | −0.129, 0.421 | 0.43 |
| Body mass | 0.131 | 0.117 | −0.099, 0.361 | 0.42 |
| Lung mass | −0.083 | 0.102 | −0.283, 0.117 | 0.33 |
| Wing stroke amplitude | 0.053 | 0.087 | −0.117, 0.223 | 0.28 |
| Sex | −0.050 | 0.217 | −0.476, 0.376 | 0.25 |
| Liver mass | 0.026 | 0.102 | −0.173, 0.225 | 0.25 |

Note:
   Variables with relative importance score >0.7 are shown in bold type and were included in further analysis.

**Table 3 Model-averaged statistical results in the best model set (delta AIC < 6) correlating total lifted load of Eurasian tree sparrows (*Passer montanus*) with all selected variables (see Table 2), and with their level-two interactions.**

| Variable | Estimate | Adjusted SE | 95% CI | Relative importance |
|---|---|---|---|---|
| (Intercept) | −0.011 | 0.086 | −0.18, 0.158 | |
| Bill length | −0.262 | 0.088 | −0.434, −0.089 | **1.00** |
| Flight muscle mass | 0.611 | 0.098 | 0.419, 0.803 | **1.00** |
| Stomach mass | 0.223 | 0.085 | 0.056, 0.389 | **1.00** |
| Wing area | 0.180 | 0.089 | 0.006, 0.354 | **1.00** |
| Gut length | 0.113 | 0.091 | −0.066, 0.291 | 0.69 |
| Bill length × Flight muscle mass | 0.142 | 0.095 | −0.045, 0.329 | 0.57 |
| Flight muscle mass× Stomach mass | −0.109 | 0.103 | −0.311, 0.092 | 0.38 |
| Bill length × Stomach mass | 0.061 | 0.090 | −0.115, 0.238 | 0.26 |
| Flight muscle mass × Wing area | −0.040 | 0.085 | −0.207, 0.127 | 0.21 |
| Stomach mass × Wing area | −0.035 | 0.087 | −0.206, 0.137 | 0.20 |
| Bill length × Wing area | −0.015 | 0.127 | −0.263, 0.234 | 0.20 |
| Flight muscle mass × Gut length | 0.065 | 0.097 | −0.126, 0.255 | 0.17 |
| Bill length × Gut length | 0.057 | 0.090 | −0.12, 0.233 | 0.15 |
| Gut length × Stomach mass | −0.034 | 0.102 | −0.233, 0.166 | 0.13 |
| Gut length × Wing area | 0.011 | 0.103 | −0.192, 0.214 | 0.10 |

Note:
   Variables with relative importance score >0.7 are shown in bold type and were included in further analysis.
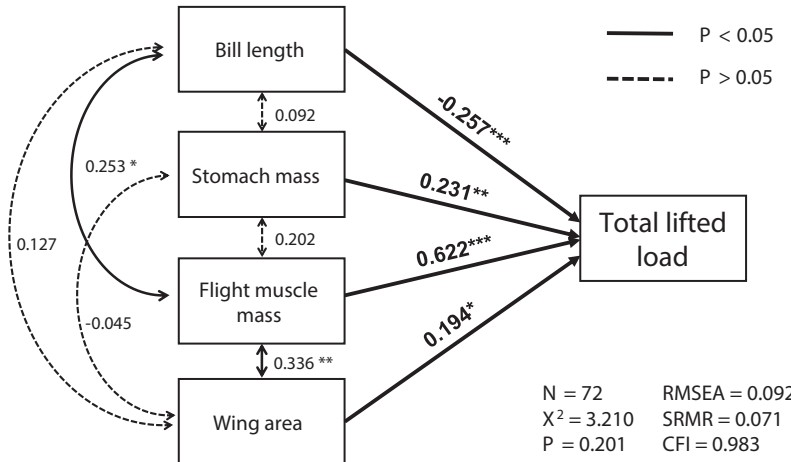

**Figure 1** **The relationships among morphological parameters and their effects on total lifted load for Eurasian tree sparrows (*Passer montanus*) in the best-fit structural equation model (SEM).** Total lifted load was positively correlated with wing area, flight muscle mass, and stomach mass, and negatively correlated with bill length. Root mean square error of approximation (RMSEA); standard root mean square residual (SRMR); comparative fit index (CFI). *$p < 0.05$; **$p < 0.01$ and ***$p < 0.001$.

The best SEM consisting of the four most important variables, including flight muscle mass, bill length, wing area and stomach mass, was well-fitted ($\chi^2 = 3.21$, $df = 2$, $p = 0.201$; RMSEA = 0.092, SRMR = 0.071, CFI = 0.983; see Fig. 1). Total lifted load was positively correlated with sizes of the flight muscles and wings, and with stomach mass, but was negatively correlated with bill length (Table S1; Fig. 1). We also found masses of the flight muscle to be positively correlated with both bill length and stomach mass (Table S2; Fig. 1).

## DISCUSSION

Many external morphological and internal anatomical measurements are positively correlated with maximum load-lifting performance in Eurasian tree sparrows (Table 1), but the strongest correlates are a greater flight muscle mass, larger wings, heavier stomach and a shorter bill (Table 3), all of which are independent of sex. Interspecifically for birds, wingbeat frequency and wing stroke amplitude are key flight kinematic variables for production of aerodynamic power output, along with larger wings (*Chai & Dudley, 1999*; *Hedenström, 2002*; *Altshuler et al., 2015*). By contrast, we here determined that wingbeat frequency and wing stroke amplitude were not strong intraspecific determinants of maximum load-lifting capacity. Variation among individuals in both wing stroke amplitude and wingbeat frequency during maximum load-lifting was small (i.e., less than 3% and 10%, respectively; see Table 1). These results are similar to previous findings in Eurasian tree sparrows showing that wing stroke amplitude during maximum load-lifting did not vary across elevational gradients (*Sun et al., 2016*). Wingbeat frequency did not affect the maximum load-lifting capacity for the lowland populations studied here but did increase in intraspecific comparisons of populations across an altitudinal gradient (*Sun et al., 2016*). Wingbeat frequency is thus a potential determinant of maximum

load-lifting capacity that can vary interspecifically among birds, and that may covary with other morphological traits such as body mass and wing area, but that is relatively invariant within individual populations of these free-living passerines during maximum load-lifting flight. Other flight kinematic parameters, such as downstroke: upstroke ratio, wing angle of attack, and stroke plane angle, may also vary among individuals in maximum load-lifting and warrant further investigation.

In volant taxa, flight muscle mass is an important feature contributing to maximum force production (*Plateau, 1865*; *Marden, 1987*). Similarly, maximum load-lifting ability (and likely power production as well) in Eurasian tree sparrows is positively influenced by more massive locomotor muscles, but also by wing area (Fig. 1; Table 3; *Lind & Jakobsson, 2001*). In general, sustained power production by flight muscle can be constrained by interactions between oxygen supply, substrate availability and muscle demand from other physiological systems, e.g., the digestive organs or respiratory and circulatory systems (*Suarez et al., 1997*; *Nespolo et al., 2018*). However, flight performance during maximum load-lifting by sparrows showed only weak correlations with the sizes of the heart, lung, gut and liver, but did show a positive correlation with stomach mass (see Table 3; Fig. 1). In short-duration flights, oxygen and energy supply may not limit such performance, which may be largely anaerobic in character (*Altimiras et al., 2017*). Alternatively, aerodynamic force production by the wings can directly constrain whole-animal vertical load-lifting (*Chai, Harrykissoon & Dudley, 1996*), as opposed to limits on power production by the flight muscle. Eurasian tree sparrows are non-migratory and also are a human commensal species (*Del Hoyo et al., 2017*) and as such are much less engaged in substantial lipid loading or digestive tract reduction (as characterizes many long-distance migrants; e.g., *Piersma & Gill, 1998*). Larger digestive organs may thus indirectly correlate with better burst flight performance in non-migratory avian species. Shorter bills were correlated with higher maximum load-lifting capacity in Eurasian tree sparrows when effects of variable flight muscle mass were incorporated (Table 3; Fig. 1), which may reflect unmeasured features of foraging behavior on flight ability, such as differential foraging strategies associated with bill size and matched by changes in flight performance. Bill size also influences heat transfer capacity in some birds (*Ryeland, Weston & Symonds, 2017*; *Tattersall, Arnaout & Symonds, 2017*) and can limit the suitability of prey items and thus foraging styles (*Cruz et al., 2001*), so that multiple aspects of bill size may be under selection.

Lift production from the wings of Eurasian tree sparrows may increase with aspect ratio, as in hummingbirds (*Kruyt et al., 2014*), but a detailed aerodynamic analysis of sparrow takeoff relative to wing design is not available. Although the Eurasian tree sparrow is typically thought of as a sexually monomorphic species, females did have shorter wings and a reduced wing area relative to males (*Monus et al., 2011*; *Sun et al., 2016*, *2017*). However, sex was not a determinant of maximum load-lifting capacity, indicating that female sparrows have comparable flight performance relative to males in spite of their smaller wings. How natural selection has enabled females to achieve similar lifting ability is intriguing and warrants further investigation.

## CONCLUSIONS

Maximum load-lifting flight performance of individual Eurasian tree sparrows was correlated with multiple morphological factors, including flight muscle mass, wing and bill lengths, wing area and stomach, but was unrelated to sex, various internal anatomical features, and measured wingbeat kinematics. Hypothesized effects of the sizes of flight muscles and wings on maximum load-lifting capacity were confirmed. Kinematic features (wingbeat frequency and wing stroke amplitude) showed no such intraspecific effects, whereas a larger bill and a smaller stomach compromised flight performance. Overall, the characterization of individual variability in flight performance in a free-living passerine indicates subtlety of interactions among multiple morphological features, some of which differ from those that have been identified interspecifically among birds. Eurasian tree sparrows are also social flockers and selection on escape performance from the ground may depend in part on group response to a perceived threat. The extent to which vertical accelerations are used in this behavior, and the extent to which other aspects of maneuverability (e.g., rotational ability) are important remain to be investigated. As a human commensal and hyperabundant bird species across the Eurasian continent, such effects are readily amenable to future study under field conditions.

### Funding

This work was supported by the National Natural Science Foundation of China (NSFC) (31672292) to Dongming Li and NSFC (31770445) to Yuefeng Wu. The funders had no role in study design, data collection and analysis, decision to publish, or preparation of the manuscript.

### Grant Disclosures

The following grant information was disclosed by the authors:
National Natural Science Foundation of China: 31672292 to Dongming Li.
National Natural Science Foundation of China: 31770445 to Yuefeng Wu.

### Competing Interests

The authors declare that they have no competing interests.

### Author Contributions

- Yang Wang performed the experiments, analyzed the data, prepared figures and/or tables, authored or reviewed drafts of the paper, approved the final draft.
- Yuan Yin performed the experiments, prepared figures and/or tables, approved the final draft.
- Shiyong Ge performed the experiments, prepared figures and/or tables, approved the final draft.
- Mo Li performed the experiments, prepared figures and/or tables, approved the final draft.
- Qian Zhang performed the experiments, prepared figures and/or tables, approved the final draft.

- Juyong Li performed the experiments, prepared figures and/or tables, approved the final draft.
- Yuefeng Wu contributed reagents/materials/analysis tools, prepared figures and/or tables, approved the final draft.
- Dongming Li conceived and designed the experiments, analyzed the data, contributed reagents/materials/analysis tools, authored or reviewed drafts of the paper, approved the final draft.
- Robert Dudley conceived and designed the experiments, authored or reviewed drafts of the paper, approved the final draft.

## Animal Ethics

The following information was supplied relating to ethical approvals (i.e., approving body and any reference numbers):

All protocols were approved by the Ethics and Animal Welfare Committee (No. 2013-6) and by the Institutional Animal Care and Use Committee (HEBTU2013-7) of Hebei Normal University.

## Field Study Permissions

The following information was supplied relating to field study approvals (i.e., approving body and any reference numbers):

Scientific collection permits (Hebei Forestry No. [2017]9) were obtained from the Department of Wildlife Conservation (Forestry Bureau) of Hebei Province, China.

## Data Availability

Raw data and code are available as Supplemental Files.

## Supplemental Information

Supplemental information for this article can be found online at http://dx.doi.org/10.7717/peerj.8048#supplemental-information.

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
