# Peer review of "Limits to load-lifting performance in a passerine bird: the effects of intraspecific variation in morphological and kinematic parameters"

_PeerJ, doi:10.7717/peerj.8048_

## Round 0.1 · original submission · Major Revisions

Three reviewers have now delivered their recommendations on this paper. As you'll see, they see value in the dataset and can perceive a broadly useful paper to come out of it, but have some major objections to how the study was framed (e.g. hypotheses) and why some parameters were chosen (e.g. theoretical basis for morphology-function relationships) as well as details of statistical analyses and more. These are constructive critiques that will improve the paper but it will require substantial revisions and re-review, with a detailed point-by-point rebuttal. One major change that may prevent further delay in re-review is to cull those data from the analysis that have no clear a priori expected relationship with flight biomechanics, and thereby simplify the study while making it more biologically realistic. I hope you agree that the paper will emerge much stronger for it -- and we would like to have it as a contribution to the literature and to PeerJ.

Reviewer 1 ·

Basic reporting

Specific comments follow.

Experimental design

Specific comments follow.

Validity of the findings

Specific comments follow.

Additional comments

L4: What led to this expectation?
L76: The authors should either present specific results from other studies, or theoretical reasons for this prediction.
L80: Delete “flight”. You may want to frame this in terms of mass-specific and marginal power.
L113: How would one camera capture amplitude? Was it perfectly orthogonal to the stroke plane at all body positions?
L116: What is “interaction frequency?”
L132: 2R^2/S
L133: The prediction that maximum load lifted will be correlated with wing loading is not really a prediction; with wing loading calculated as maximum load lifted divided by wing area, it must be correlated. Why was ‘load lifted’ included in wing loading? Why not just a pure morphological parameter body mass/area?
L150: “…[greater] flight muscle masses..”
L154: Again, not sure max wingloading is purely dependent; it contains the independent variable wing area.
L160: Why an AIC of <6? Ref.
L227: “in [flights of short duration], oxygen and….”
L240: How might thermoregulation and prey selection influence burst lifting performance?
L243: Are your frame rates insufficient to determine accelerations resulting from leg push before the first downstroke? You should be able to determine roughly when the resulting upward velocity is lost. Wind resistance would be small.
L256: “[correlated with….”] “Influenced” denotes causality.
Legends in Figure 1 A and B are reversed.
The correlation of bill length: Insignificant correlation with total lifted load. Yet it leaps to significance in the model averaged results. Why?
Incidentally, searching for correlations of something so difficult to precisely measure as bill length (averaged for the two sexes at 8.5 sd <0.5 mm!) invites spurious correlations.
The best part of this study is the finding that frequency and stroke amplitude were not correlated with load lifting. If true, the only physical parameter left that could explain their greater performance is downstroke force, which would indeed correlate with pectoralis mass.
I think there are some interesting data here, and I’d hate to have this much work be for naught. The authors should trim this down and focus on those morphological variables that could possibly affect flight performance, and keep them simple and physically relevant. If more precise kinematics are available, the authors should report them.

Reviewer 2 ·

Basic reporting

Please see below my general comments...

Experimental design

Please see below my general comments...

Validity of the findings

Please see below my general comments...

Additional comments

General Comments
Please find below my inquiries and suggestions that must be addressed before I recommend this paper for publication.

I) GENERAL COMMENTS AND SUGGESTIONS

A) Climbing Rate During Ascending
During trials, sparrows seem to be ascending at different rates during load-lifting experiments (from Supplementary Videos in Sun et al. 2016, doi: 10.1242/jeb.142216). Moreover, climbing flight duration seems longer than the very brief ‘hovering’ (Could you please add some supplementary videos?). Thus, the total mechanical power must depend on the climbing speed (i.e. the rate of kinetic energy, see Berg and Brewer 2008, doi: 10.1242/jeb.010413) and not just on the maximal induced power (i.e. maximum lift). For example, consider two birds lifting the same weight, the one climbing at higher speed will experience higher costs.
Without knowing the climbing speed it is hard to understand the “Limits to load-lifting performance in sparrows”. What is the speed rate during ascending? What is the actual ‘hovering’ duration? Please consider including this in your analysis.


B) Jumping from the ground

Bird are using their legs to gain a large initial speed which can impact maximal lift production.
Can you report what is the speed after the birds are not touching the ground? How related is it with the tarsus length?

C) Tail morphology and kinematics
It has been reported that tail in passerines plays an important role during flight (see Su et al. 2012, J R Soc Interface, doi:10.1098/rsif.2011.0737). Can you report relevant tail kinematics and morphology using your videos and include them in your analysis?

D) Down-stroke Duration
In songbirds, aerodynamic forces are mostly generated during down-strokes. Can you include downstroke duration in your analysis?

E) Wall effects in a closed plexiglass chamber
Were birds ascending in the center of the chamber or some were ascending close to the walls? If so, ave Dobirds flying closer to the walls have more load-lifting capacity?



II) SPECIFIC COMMENTS

Line 105: "Briefly, a thread with different plastic beads (each approximately 1.0 g in mass)" .
-Did you include the thread mass in your calculations? In video 2 from Sun et al. 2016 seems like the thread was 1 meter long before the first plastic bead was lifted by the bird.

Line 159: "To account for model selection uncertainty, model-average estimates of variable coefficients were computed using the ‘best model set’, defined as the set of models for which delta AIC was less than 6".
-What was the criterion to choose the AIC value?

-Perhaps you can include the following papers on your discussion regarding wing loading.
Lind 2001.Escape flight in moulting Tree Sparrows (Passer montanus).Functional Ecology 15, 29 – 35
Lind and Sven Jakobsson 2001.Body building and concurrent mass loss: flight adaptations in tree sparrows (DOI: 10.1098/rspb.2001.1740)

Reviewer 3 ·

Basic reporting

Some fundamental literature references are missing, e.g. Marden (1987), J Exp Biol
The hypotheses should be better rooted in biomechanics. I did not understand the rational for including many of the variables, such as bill size and internal organ sizes, and these need explanation.

Experimental design

This is a statistical approach to a biomechanics problem, and many variables measured and included in statistical models don't seem to be well motivated. More information is needed about the flight assay, nit simply referring to another paper (that I haven't read).

Validity of the findings

There is a lack of connection between the results and biomechanics (aerodynamic) relationships.

Additional comments

This paper investigated the sprint load lifting capacity of tree sparrows, which were exposed to a load-lifting assay. This assay consists of take-off flight with an attached thread of beads spaced along the thread, and the asymptotic load lifting capacity is measured. This was then analysed statistically against a great number of measures taken on the birds, including many internal organs after that the birds had been sacrificed. Why some of these measured were thought to influence load-lifting capacity was not always clear and not well motivated. I would have preferred a more hypothesis-driven approach where relationships based on aerodynamics are used to derive qualitative/quantitative predictions. It is nowhere mentioned that the assay refer to anaerobic muscle work, while some of the relationships referring to internal organ size (liver, intestine) should rather be expected to influence aerobic flight capacity. Also, what is the rational to belive that bill length will influence load-lifting capacity. Wing area was taken as the area of one wing doubled, while the convention in animal flight research is to include the area of the body between the two wings (there is no motivation for excluding this area). There are also confusion about some measures, where for example wing-related morphology is sometimes area and sometimes length, and even “wing size” is used without defining what it is.
Line 189 That maximum wing loading is related with body mass is a direct consequence from its definition (mg/wing area), and need no statistical analysis.
Line 233 why would migratory species (taxa) have more pronounced interactions among “physiological systems”?
There are some literature omissions, for example Marden (1987) , J Exp Biol.
Line 256 Please note that by the statistical analysis you could not infer that a certain factor is “influencing” load lifting, but only that there is a correlation.
Line 261 “in contrary to our initial hypothesis, size of heart, lungs and digestive organs were unrelated to maximum load-lifting capacity,..”. I did not understand the rational for hypothesising such a relationship in the first place.

---

## Round 0.2 · Minor Revisions

We have obtained two reviews that are constructive and urge moderate revisions, mainly to clarify the methods of data analysis (e.g. wingbeats vs. recording at 250 Hz) and to think more carefully about interpretations, as well as prior literature that is relevant to the paper. Please revise, track changes and provide a detailed Response. If the revisions are attentive then we may not require further review. Thank you!

Reviewer 4 ·

Basic reporting

My detailed comments point out where they fall short in this respect.

Experimental design

Nothing to add. I still don't understand why they used such a small flight arena, but apparently, according to their reply to other reviews, birds habituated to humans can fly just fine in a small box?

Validity of the findings

I didn't check the supplementary data, so maybe it's there, but they could easily provide a table of all of the primary data. People could then run any statistical model they saw fit.

Additional comments

This study contains a large sample of short burst load lifting performance data for a single species of bird from a single location, with a set of morphological and kinematic variables. The premise of the study is that some body features are expected to increase performance whereas others should reduce performance.

This version of the manuscript is improved over a previous one I reviewed for a different journal. There is now a hypothesis and Fig 1 summarizes the main data. As my comments show, I have some issues with the way they have posed and reflected on their hypotheses, and I’ve offered some constructive comments for them to consider.

A problem that still remains is that they only examined the total load lifted (body plus added mass). This is a fine start, but to better understand the result, it would be helpful if they extended the analyses to evaluate some sub-hypothesis. For example, the results are consistent with prior literature showing that burst flight performance depends strongly on the mass of flight muscles and, and that the added load capacity depends strongly on the ratio of flight muscle mass to total body mass. Between species, the total load relationship scales as flight muscle mass^1.0 or very nearly so (isometry). We know less about this scaling within species. Might they show their result? Using that result, they can evaluate the residual variation to see if e.g. a larger stomach or smaller beak can explain a significant portion of the residual variation in load lifted. Clearly this is different from their approach that puts all variables on equal footing and has flight muscle mass counted twice (by its inclusion in body mass), but flight muscle mass was the expected key variable prior to their analysis and is the key variable after their analysis. Hence, why not use an analysis in which it is the major causational variable and then show how the other variables contribute? Doing so may provide much clearer insights. As an example, consider total load and body mass. More massive birds likely have larger flight muscles, which increase total load capacity. But larger body mass can result from many different tissues. Anything they can do to tease apart the size dependencies within their data would be helpful. Perhaps that is what their only figure aims to do but I find it uninformative.

For example, a specific hypothesis they could address is whether a small beak is an indicator of a relatively small non-muscle body mass. Larger muscles in a bird with a relatively small beak might indicate large muscles in relation to skeletal size of the anterior central body core. Similarly, if they sum standardized variables for the digestive mass/size (stomach mass, intestine length, liver mass), would they get a stronger result that better supports their (not yet very clearly stated) hypothesis that gathering nutrition co-varies the energy and/or condition of flight muscles and thereby affects short-burst performance?

Specifics:

L. 24. I’m not convinced that the paper reveals how/why these parameters are “constrained”, so I find this opening sentence to be a bit misleading. “Interrelated” would be a more accurate term for the approach this study uses.

L. 28. As multiple reviewers have pointed out, any prediction about organ size needs to consider the difference between short bursts of anaerobic effort vs. sustained effort. Short burst performance might predictably be related to smaller hearts, lungs and stomachs, i.e. a bird that is nearly all muscle+wing and little else. The paper they cite on the small heart size of Tinamous is an excellent example of authors distinguishing between burst and sustainable performance. Similarly, I recall a paper showing that palatable butterflies that need to evade predatory birds have larger flight muscles and their gut and ovarian mass are reduced compared to chemically protected butterflies that have little or no need to evade birds.

L. 28 There are two other problems with this sentence. First, if bigger muscles, hearts, stomachs, and lungs are predicted to be favored, then why not intestines, eyes, blood, brains and beaks – all of which could contribute to the gathering and assimilating of food to provide energy for flight? Second, if bigger is better for all of these things, how can you simultaneously predict “relatively” smaller body size? Relative to what? The only way this makes sense is if body parts unrelated to supporting mechanics and energetics of flight are reduced. I suppose this would have to be bones, feathers, kidneys and reproductive organs, but certainly bones and feathers support muscles and aerodynamic forces, so perhaps that list is shorter. The broader point is that the hypothesis as posed is poorly formed and needs to consider the physiological role of different tissues/organs in relation to different modes of flight (burst vs. sustained).

L. 59: What does “upregulated” mean in this context? Do you mean hypertrophied?

L. 62: “As major metabolic engines, sizes of heart and lung also underpin capacity for both burst power and endurance flight performance (see Bishop and Butler, 1995; Tobalske et al., 2003; Wright et al., 2014).” The Tobalske paper is about power vs. speed relationships; the words “heart” and “lung” do not appear in that paper.

More importantly, why expect heart and lung effects in an experiment in which flight bouts lasted only seconds? Is this enough time for O2 to get from the lung surface to the mitochondria and thereby affect burst performance?

L. 78: There is no reason to specify a method in these references or otherwise differentiate them as it has been shown that the two methods produce indistinguishable results.

L. 91: “including the body”. Where in the analyses is body mass evaluated? I don’t see an allometric equation relating total load or added load to unladen body mass. The positive correlation between total load lifted and body mass in not informative on this point.

L. 141: methods describing flight muscle dissection: The pectoralis mass shown in Table 1 strikes me as too low (12.8% of body mass). Might you have done the dissection on only one side and are reporting only half of the total pectoral muscle mass?

L. 167: cite the method used for centralizing and standardizing

L. 221: For reasons I described in a previous review of an earlier version of this paper submitted to a different journal, you should also analyze the product of wingbeat frequency and amplitude, or at least state explicitly why your video method does not allow you to resolve these variables at the critical time point (i.e. I now understand from your responses to other reviews that you may be generally missing wingbeat amplitude at the max load lifted). The reader should know the limitations of the data and understand why this product is not being presented, particularly since you are reporting the two component variables of this product.

L. 232. Why does this sentence begin with “In hummingbirds”? This has been known for birds in general, including Passer, since 1987. This strikes me as an egregious example of self-citing.

L. 247: “aerodynamic force production by the wings can constrain whole-animal vertical load-lifting”. This is always true; force output is limited by force production. The informative thing to say is why force production during short bursts is NOT limited by aerobic capacity.

L. 252: “Larger digestive organs may thus enhance burst flight performance in non-migratory avian species.” Finish the thought. You earlier cited literature suggesting that digestive capacity may affect the condition of the flight muscles. Use this possibility to say why a continuous-time process (protein and energy assimilation) may relate to motor performance, even during short bursts, perhaps by affecting the glycogen and protein content of the flight muscles. Thinking hard about your results in this fashion may help you devise you next project and/or get your next grant.

L. 253: I encourage expanding a bit your discussion of bill length effects. You could start by estimating the mass difference of different size bills, which will undoubtedly by trivial. This sets you up to say that it instead may reflect foraging differences. Perhaps you are familiar with the famous studies of Galapagos finches showing that drought and wet years affect the abundance of seeds of different sizes and the fitness of finches with different beak sizes. Perhaps you know something about what seeds were abundant in the days & weeks preceding your study. You could speculate that, if foraging success underlies the beak size result, that it could change seasonally as different plants produce seed and this species’ diet changes. Perhaps you know something about beak sizes in different populations and could suggest that there is gene flow for which beak size may be a marker and that genetic differences between populations may include different burst performance capability, in which case it may not be beak size per se that is affecting load lifting but rather the associated genetic differences. Biology always challenges us to think broadly!!

L. 278: We certainly know that burst accelerations are used by birds such as sparrows to evade predatory birds. This can be observed every day at the bird feeders in my yard when Cooper’s Hawks come by looking for a meal. You needn’t state complete lack of knowledge/insight here. See for example https://royalsocietypublishing.org/doi/10.1098/rspb.1996.0244,
https://jeb.biologists.org/content/218/2/212?version=meter+at+null&module=meter-Links&pgtype=article&contentId=&mediaId=&referrer=&priority=true&action=click&contentCollection=meter-links-click, https://www.jstor.org/stable/4163245?seq=1#metadata_info_tab_contents,
https://academic.oup.com/auk/article/117/4/1034/5147010,
https://www.sciencedirect.com/science/article/pii/S0022519304002899,
and others by Lind and co-authors.

Reviewer 5 ·

Basic reporting

More information is needed about the methods for obtaining wingbeat amplitude, and the previous recommendation by a reviewer to report downstroke ratio (percent time in downstroke) should be revisited.

In lines 116-118 it is written that the top camera (250 Hz) was used to measure amplitude, but on line 122 it is written that wing-stroke amplitude was derived from lateral video (50 Hz). Looking at the supplemental videos, I am concerned that there is excessive blur due to the use of low shutter speeds. First, it is important to clarify which camera view was used to measure amplitude. Then the specific methods, more clear than in the Sun et al. 2016 JEB paper, would be helpful. Was the maximum excursion of the wing assessed from shoulder to the tip of the primaries. If so, which primary tip? The stroke planes do not seem to be completely parallel to global horizontal. In general, what is the stroke plane angle relative to horizontal, and what is the estimated error involved in using only one camera to calculate amplitude given this average stroke-plane angle?

A previous reviewer suggested reporting downstroke ratio (percent time spent in downstroke within a wingbeat), but the authors replied that the measure is not possible given their frame rates. 250 Hz sampling seems sufficient for this purpose, and it is not clear how the authors can obtain wingbeat amplitude without being able to quantitatively discriminate between downstroke and upstroke.

Experimental design

No comment

Validity of the findings

Since wingbeat kinematics are reported to not correlate significantly with performance, it is vital to improve the description of how wingbeat amplitude was measured.

Additional comments

You report interesting and worthwhile data that explore issues related to your 2016 study.

---

## Round 0.3 · accepted · Accept

The reviewers feel that more of their comments could have been taken into account in revision but that the paper is satisfactory now. There is still an opportunity to take their comments on board (from prior reviews + comment on downstroke % repeated here) and improve the MS but that is your decision. Thank you for your patience through the review process.

Reviewer 4 ·

Basic reporting

no comment

Experimental design

no comment

Validity of the findings

no comment

Reviewer 5 ·

Basic reporting

This is satisfactory.

Experimental design

This is satisfactory

Validity of the findings

This is satisfactory.

Additional comments

I appreciate your clarification of your methods for measuring wingbeat amplitude. It is still unclear to me why downstroke (%) is not obtainable, but I don't view this variable as being critical to your experimental design and interpreting your results.